# Phenotypic, Technological, Safety, and Genomic Profiles of Gamma-Aminobutyric Acid-Producing *Lactococcus lactis* and *Streptococcus thermophilus* Strains Isolated from Cow’s Milk

**DOI:** 10.3390/ijms25042328

**Published:** 2024-02-16

**Authors:** José Alejandro Valenzuela, Lucía Vázquez, Javier Rodríguez, Ana Belén Flórez, Olga M. Vasek, Baltasar Mayo

**Affiliations:** 1Departamento de Microbiología y Bioquímica, Instituto de Productos Lácteos de Asturias (IPLA), Consejo Superior de Investigaciones Científicas (CSIC), Paseo Río Linares s/n, 33300 Villaviciosa, Spain; josea.valenzuela@hotmail.com (J.A.V.); lucia.vazquez@ipla.csic.es (L.V.); javier.rodriguez@ipla.csic.es (J.R.); abflorez@ipla.csic.es (A.B.F.); 2Biotecnología Microbiana para la Innovación Alimentaria, Instituto de Modelado e Innovación Tecnológica-Universidad Nacional del Nordeste (CONICET-UNNE), Campus UNNE, Corrientes 3400, Argentina; omvasekk@yahoo.com.ar; 3Instituto de Investigación Sanitaria del Principado de Asturias (ISPA), Avenida de Roma s/n, 33011 Oviedo, Spain

**Keywords:** *Lactococcus lactis*, *Streptococcus thermophilus*, starters, diary cultures, dairy products, cheese, yogurt, milk fermentation

## Abstract

Gamma-aminobutyric acid (GABA)-producing lactic acid bacteria (LAB) can be used as starters in the development of GABA-enriched functional fermented foods. In this work, four GABA-producing strains each of *Lactococcus lactis* and *Streptococcus thermophilus* species were isolated from cow’s milk, and their phenotypic, technological, and safety profiles determined. Genome analysis provided genetic support for the majority of the analyzed traits, namely, GABA production, growth in milk, and the absence of genes of concern. The operon harboring the glutamate decarboxylase gene (*gadB*) was chromosomally encoded in all strains and showed the same gene content and gene order as those reported, respectively, for *L. lactis* and *S. thermophilus*. In the latter species, the operon was flanked (as in most strains of this species) by complete or truncated copies of insertion sequences (IS), suggesting recent acquisition through horizontal gene transfer. The genomes of three *L. lactis* and two *S. thermophilus* strains showed a gene encoding a caseinolytic proteinase (PrtP in *L. lactis* and PrtS in *S. thermophilus*). Of these, all but one grew in milk, forming a coagulum of good appearance and an appealing acidic flavor and taste. They also produced GABA in milk supplemented with monosodium glutamate. Two *L. lactis* strains were identified as belonging to the biovar. diacetylactis, utilized citrate from milk, and produced significant amounts of acetoin. None of the strains showed any noticeable antibiotic resistance, nor did their genomes harbor transferable antibiotic resistance genes or genes involved in toxicity, virulence, or pathogenicity. Altogether these results suggest that all eight strains may be considered candidates for use as starters or components of mixed LAB cultures for the manufacture of GABA-enriched fermented dairy products.

## 1. Introduction

Gamma-aminobutyric acid (GABA), a non-protein amino acid synthesized from glutamic acid, acts in the human central nervous system as an inhibitory neurotransmitter affecting nerve impulses. GABA reduces stress and improves the onset and duration of sleep; it thus acts as a tranquilizer [1]. In addition, GABA modulates kidney function and lowers blood pressure [2]. Some of these effects have already been demonstrated in randomized clinical trials [1,3,4], strongly supporting the idea that GABA can have a beneficial influence on human health and well-being. These properties of GABA have attracted much interest in terms of developing GABA-enriched functional foods [5].

The transformation of glutamic acid into GABA is carried out by the enzyme glutamate decarboxylase (GAD), using pyridoxal phosphate (the active form of vitamin B6) as a cofactor [6]. Caseins have a high concentration of glutamic acid, and thus, the GABA precursor is therefore abundant in milk. GABA production in fermented dairy products could be achieved if free glutamic acid plus microorganisms (from starter cultures, adjuncts, or contaminating microbiota) with GAD activity are present [7,8]. Indeed, the use of GABA-producing lactic acid bacteria (LAB) as a starter and/or adjunct culture, allows for the production of GABA-enriched functional fermented dairy products [9,10]. GAD activity has been detected in strains of different LAB species, including *Levilactobacillus brevis* (formerly *Lactobacillus brevis*), *Lacticaseibacillus paracasei* (formerly *Lactobacillus paracasei*), *Lactococcus lactis*, *Streptococcus thermophilus*, and many others [11,12]. GAD activity seems to provide microorganisms with greater resistance to low pH, such as that found in most fermented foods [13]. In certain LAB, the decarboxylation of glutamic acid into GABA coupled with the exchange of these compounds through the membrane provides cells with extra energy in the form of ATP that can be used for enhanced growth [13].

*L. lactis* and *S. thermophilus* are the most economically important LAB species and are used worldwide as starters in cheese making and yogurt manufacturing, respectively [14]. However, only certain strains of these two species show GAD activity and are therefore capable of GABA production [8,10,15,16]. The enzyme in *L. lactis* is encoded only by the gene *gadB*, which forms part of the *gadRCB* operon that includes two upstream genes coding for a GABA–glutamate antiporter (*gadC*) and a transcription regulator (*gadR*) [17]. The *gadRCB* operon in *L. lactis* is recognized as part of an acid stress regulon And inactivation of either *gadB* or *gadC* results in reduced cell survival at low pH [18]. Genes equivalent to *gadB* and *gadC* in *L. lactis* are also found in *S. thermophilus* [7]. However, compared to *L. lactis*, the order of the genes in the operon is reversed and the gene encoding the regulator is absent [19]. Further, in *S. thermophilus* the GAD operon is flanked by transposon- and insertion sequence-derived sequences, suggesting a horizontal acquisition [19]. In addition to GABA production, the confident use of *L. lactis* and *S. thermophilus* strains as starters requires strains to be subjected to exhaustive biochemical and genetic characterization to ensure that they comply with the demanded technological and safety requirements [20].

Aiming at identifying GABA-producing starter candidates of *L. lactis* and *S. thermophilus*, a set of strains of these species isolated from raw milk were surveyed for GABA production. Among the higher producers, a set of eight strains (four *L. lactis* and four *S. thermophilus*) was subjected to a battery of phenotypic tests to determine their main technological and safety properties, followed by genome sequencing to reveal the basis of many of their key traits, including the synthesis of GABA. This manuscript reports on the phenotypic and genetic features that allowed us to propose the strains as conventional and functional starter cultures for dairy.

## 2. Results

### 2.1. Molecular Identification and GABA Production

The molecular identification of 119 isolates from PCA plates incubated at 32 °C (one isolate was lost during recovery) proved lactococcal species to be the majority population in raw milk, accounting for *circa* 35% of the isolates, including *L. lactis* (26 isolates), *Lactococcus raffinolactis* (8 isolates), and *Lactococcus garvieae* (7 isolates). Similarly, among 117 colonies recovered from the GLM17 plates incubated at 42 °C (three isolates were lost during purification or recovery), the majority of the isolates were identified as *S. thermophilus*, accompanied by a few isolates of *Staphylococcus epidermidis*, *Lactobacillus delbrueckii*, *Streptococcus gallolyticus* subsp. *macedonicus*, and *Streptococcus equinus*. These isolates accounted for the species withstanding pasteurization and high incubation temperature. Thus, under the specific isolation conditions of this study, *L. lactis* (21.8%) and *S. thermophilus* (41.8%) proved to be, respectively, the majority of mesophilic and thermophilic LAB species in milk.

Since *L. lactis* and *S. thermophilus* isolates were obtained from every milk sample, to avoid testing replicates, an isolate of each of the species per sample was assayed for GABA production in GM17 (for *L. lactis*) or GLM17 (for *S. thermophilus*) broth supplemented with 5 Mm monosodium glutamate (MSG). GABA was detected in the supernatant of all cultures, although tiny amounts were measured in some isolates. Six *L. lactis* and eight *S. thermophilus* isolates converted most of the MSG precursor into GABA (>4.0 mM; >80% conversion). These 14 isolates were subjected to RAPD and rep-PCR typing. Taking into account the repeatability of the method (93%), the combined fingerprinting results suggested the presence of six different strains for each species (Appendix A). Among those, four genetically unrelated strains each of *L. lactis* (Lc 5.5, Lc 14.4, Lc 19.3, and Lc 21.1) and *S. thermophilus* (St 8.1, St 9.1, St 18.1, and St 21.1) were selected for the characterization of relevant biochemical, technological and safety properties, as well as for genome sequencing and analysis.

### 2.2. Biochemical and Technological Characterization

The carbohydrate fermentation profile of the strains, as determined by the API-50 CHL system, matched those of their respective species (Appendix A). The *L. lactis* profile ranged from 13 to 21 carbohydrates. Lc 21.1 utilized the smallest number of compounds, which is typical of well-adapted-to-milk (“domesticated”) lactococcal strains. The profile of the *S. thermophilus* strains included the fermentation of 3–5 carbohydrates, a short but typical sugar-utilization profile for this species.

Species-specific profiles for enzyme activities, assayed using the API-ZYM system, were also scored for the *L. lactis* and *S. thermophilus* strains (Appendix A). The strains of the two species showed strong leucine aminopeptidase and phosphohydrolase (against naphthol-AS-BI) activities. The *L. lactis* strains showed strong acid phosphatase activity and moderate cysteine aminopeptidase activity. In general, the *S. thermophilus* strains showed strong β-galactosidase activity, moderate valine aminopeptidase and esterase activities, and weak lipase activity. With variable enzymatic levels, all *S. thermophilus* strains returned a positive urease test.

Two strains of each of the species (Lc 5.5 and Lc 14.4, and St 8.1 and St 18.1) grew well and coagulated the milk by 12 h of incubation (pH 4.43–4.70), forming a firm coagulum of good appearance and pleasant acidic flavor and taste. In contrast, the pH of the milk at this time was ≥6.0 for all other strains, and none of these clotted the milk even after 24 h of incubation. Minor differences in acidification were observed when the milk was supplemented with 10 mM MSG (Figure 1). GABA production in supplemented milk was primarily associated with growth in this medium (Figure 1); strains that did not grow in milk were not able to synthesize GABA in this medium. However, GABA production by *S. thermophilus* was retarded compared to that recorded for *L. lactis*. At 12 h of incubation, only *L. lactis* Lc 5.5. and Lc 14.4 produced significant amounts of GABA, while at 24 h, three *L. lactis* (Lc 5.5, Lc 14.4, and Lc 19.3) and two *S. thermophilus* (St 8.1 and St 18.1) strains did so. In laboratory media, the presence of MSG caused a small increase in the final pH of the cultures (0.23 ± 0.06). However, this effect was not observed in milk. Finally, under the conditions of this study, none of the strains produced GABA in MSG-supplemented fecal homogenates (the growth of the strains in this system was not recorded).

The strains produced and consumed different amounts of organic acids and sugars. However, no clear-cut species-specific profiles were observed (Table 1). In *L. lactis*, major differences between the strains were associated with the utilization of citrate. Strains using this compound (Lc 14.4 and Lc 21.1) produced large amounts of acetic acid. In addition, strains Lc 5.5 and Lc 14.4 produced moderate amounts of formic and succinic acids. In *S. thermophilus*, the profiles for sugar fermentation and the production of organic compounds seemed to be mostly influenced by growth in milk. Strains St 8.1 and St 18.1 grew well and generated large amounts of lactic acid, while releasing large quantities of galactose. Strains St 9.1 and St 21.1 did not grow well in milk and, in general, their metabolic activity was rather poor. At the same time, they both released notable quantities of glucose, suggesting the premature lysis of cells.

After growing the strains in milk at either 32 °C (*L. lactis* strains) or 42 °C (*S. thermophilus* strains), 29 volatile compounds were detected (Table 2), 15 of which were not present in the starting milk. For the *L. lactis* strains, no species-specific profile was observed due to the different behavior shown by the Lc 21.1 strain; unlike its counterparts, it did not produce isovaleric acid, 4-octanone, or phenylacetaldehyde. Further, Lc 5.5 and Lc 14.4 produced 2-phenyl ethanol, isoamyl alcohol, and 5-methyl-2-phenyl-2-hexenal. Lc 14.4 produced acetoin (diacetyl cannot be detected with the solid-phase microextraction (SPME)–gas chromatography method utilized in this study), and large amounts of acetic acid. Lc 5.5 and Lc 19.3 both produced notable amounts of benzoic acid, while Lc 14.4 and Lc 19.3 produced 2,5-dimethyl-3-hexanone and 2,3-heptanodione. In contrast, with only minor differences between the strains, species-specific profiles were seen for the four *S. thermophilus* strains. Two species-specific volatile compounds were associated with *S. thermophilus*, namely, tetradecanoic acid and 2-heptanone.

Under the assay conditions, none of the strains produced bacteriocin-like inhibitory substances active against *L. sakei* CECT906^T^.

### 2.3. Safety Evaluation

The MIC values of the tested antibiotics showed the strains to be susceptible to all relevant antibiotics (Appendix A). All strains were strongly resistant to trimethoprim, a resistance intrinsic to cocci LAB. Deviating from the profile, high MIC values of rifampicin (>64 µg mL^−1^) and kanamycin (128 µg mL^−1^) were displayed by *L. lactis* Lc 21.1 and *S. thermophilus* St 9.1, respectively. Under the culture conditions assayed, none of the isolates produced biogenic amines from any of the four precursors tested (histidine, tyrosine, agmatine, and ornithine).

### 2.4. Genome Sequence and Analysis

To understand the genetic basis of GABA production—and to help ensure their safe use—all eight strains were subjected to genome sequencing and analysis. The acquired genomic data served to confirm the taxonomic identification of the strains via in silico dDDH and orhtoANI analyses. The examined strains were compared with type strains of related species, including *L. lactis* subsp. *lactis*, *L. lactis* subsp. *lactis* biovar. diacetylactis, and *L. cremoris* subsp. *cremoris*, or the *S. thermophilus* type strain and the type strains of related species (Appendix A).

These analyses confirmed the previous identification of the strains as either *L. lactis* or *S. thermophilus*. Further, *L. lactis* Lc 21.1. was identified as belonging to the biovar. diacetylactis. Although lower than those of Lc 21.1, the dDDH and orhtoANI values indicated Lc 14.4 to be slightly closer to the *L. lactis* subsp. *lactis* biovar. diacetylactis type strain (GL2^T^) than to the *L. lactis* subsp. *lactis* type strain (ATCC 19435^T^).

Table 3 shows the general features of the genomes of all the sequenced strains. Genomic analysis of the strains showed that they all carried the gene coding for the GAD enzyme (*gadB*) and its corresponding associated antiporter (*gadC*), organized in an operon-like structure (Figure 2). The assembly of the genome of the *L. lactis* strain Lc 14.4 split the operon into four contigs, impeding the determination of its structure. One contig harbored genes coding for proteins involved in plasmid replication, which strongly suggests that there are two copies of the GABA operon in the genome of this strain, of which one is plasmid-encoded. This structure may account for the assembly failure in Lc 14.4 (Figure 2A). As expected, the order of the GAD genes in *L. lactis* and *S. thermophilus* was reversed, and the open reading frames (ORFs) flanking them were different in the two species (Figure 2B). The GABA operon in *L. lactis* was flanked by chromosomal genes encoding ribonuclease HII (*rnhB*) and a potassium efflux system (*kefA*). Located at the first position, the operon in *L. lactis* also comprised the regulatory gene (*gadR*) (Figure 2A). The *S. thermophilus* operon also appeared to be encoded on the chromosome, and in the same position in all strains. However, the operon in this species lacked the gene encoding the regulator and was flanked by sequences coding for insertion elements and mobilization proteins (Figure 2B), suggesting (recent) acquisition by horizontal transfer.

The genetic makeup for lactose and galactose utilization included clusters for lactose utilization via the Leloir pathway in both the *L. lactis* and *S. thermophilus* strains, and an operon for lactose utilization via the tagatose-6-phosphate pathway in the genomes of all lactococcal strains (Appendix A). These pathways are well characterized in strains of many LAB species, including *L. lactis* and *S. thermophilus* [21].

Genome analysis of the citrate-associated genes in the two *L. lactis* strains utilizing citrate (Lc 14.4 and Lc 21.1) revealed striking differences (Appendix A). Three operons, each located in a different contig, were identified in Lc 21.1. Two operons were considered to be located on the bacterial chromosome, encoding the citrate lyase complex and the malolactic enzyme. Each of the operons harbored a gene encoding a citrate/H+ symporter-like permease, which contained mutations that might render the enzymes inactive. The third operon was found in a contig together with genes encoding plasmid-replication proteins, strongly suggesting a plasmid location.

The latter operon also contained an ORF encoding the well-described citrate-specific transporter CitP. In contrast, only the chromosomal operons encoding citrate lyase and malolactic enzyme were identified in the genome of the Lc 14.4 strain. Surprisingly, the genes coding for the corresponding associated permeases were complete and contained no mutations. Genes encoding citrate synthase and isocitrate dehydrogenase were found in the genomes of all *L. lactis* strains; these enzymes might intervene in the synthesis of citrate.

The analyzed genomes showed a complex repertoire of metabolic genes involved in the growth of the strains in milk and the formation of taste and aroma compounds (Appendix A). With only minor differences between strains, gene content species-specific profiles were established. Nonetheless, these minor differences might be critical for the rapid development and/or for the synthesis of flavor compounds in milk. In addition to the genetic machinery involved in diacetyl production from citrate, genes coding for the *L. lactis* and *S. thermophilus* caseinolytic proteinases (PrtP and PrtS, respectively) were identified in most strains, but not in Lc 21.1, St 9.1, and St 21.1. High cell densities and milk coagulation after overnight culture were only attained by the strains harboring either *prtP* or *prtS* genes, except for the Lc 19.3 strain. As expected from the biochemical characterization results, a complete urease operon consisting of three catalytic enzymes and seven accessory genes was found in all four *S. thermophilus* strains (Appendix A).

No CRISPR-Cas-associated loci were identified in the genomes of the *L. lactis* strains; these systems are uncommon in this species. In agreement with the absence of CRISPR-Cas systems, a large proportion of phage-associated genes was identified in the genome of all *L. lactis* strains (Table 3). The Phaster software associated these genes with either complete (from one to three) or incomplete (from 4 to 13) phages. In contrast, all four *S. thermophilus* strains possessed three loci showing both CRISPR repeats and genes coding for Cas proteins. Scattered in the genomes, a short series of isolated CRISPR repeats was also found in all strains except in St 8.1. Only two loci showing an incomplete set of genes of phage origin were detected in all *S. thermophilus* strains. In addition, a complete phage of 43.9 kbp was identified in strain St 21.1. PlasmidFinder identified two plasmid-derived sequences in St 9.1, while one plasmid-replication protein was detected by BV-BRC in St 8.1, St 9.1, and St 18.1 (Table 3). PlasmidFinder has been reported not to be reliable for the detection of plasmids in LAB species [22]. Despite this, the software identified plasmid sequences in all *L. lactis* strains, which agrees well with the large number of genes encoding putative plasmid-associated proteins detected by BV-BRC in all lactococcal genomes (Table 3).

Despite the absence of antimicrobial activity in the agar well-diffusion assay, several loci were identified by BAGEL 4 as putative regions encoding bacteriocin-like peptides (Blp) or ribosomally synthesized and post-translationally modified peptides (RiPPs) (Table 3). Besides several lactococcins, Lc 14.4 encoded a protein with homology to enterolysin A (53% amino acid identity). Three lactococcal strains showed a locus for the synthesis of sactipeptides (sulfur-to-α-carbon-containing peptides, also classified as Class Ic bacteriocins) (Table 3). Finally, a locus for the synthesis of linaridin (a class of linear, dehydrated peptides) was detected in the genome of Lc 19.3. For the *S. thermophilus* strains, loci involved in the synthesis of several Blp antimicrobials (Class II bacteriocins with a double-glycine leader peptide) were identified. In addition, they all contained a locus for the synthesis of streptides (a class of 20-membered cyclic peptides). Finally, St 8.1 showed a locus with the capability to encode a bacteriocin homologous to the BhtR bacteriocin (a Class I two-component lantibiotic) identified in *Streptococcus ratti* [23].

Interrogation of the genomes with the CARD database for antibiotic resistance genes detected two strict hits in the genomes of all four *S. thermophilus* strains with low homology (<33% identity) to the deduced proteins for the *vanT* and *vanY* genes of *Enterococcus* spp. These genes were annotated by BV-BRC as alanine racemase and D-alanyl-D-alanine carboxypeptidase, respectively. Neither genes nor mutations involved in antibiotic resistance were detected in the *S. thermophilus* genomes by ResFinder. The D-alanyl-D-alanine carboxypeptidase gene was also detected as a strict hit by CARD in all *L. lactis* strains. In addition, this database identified the *lmrD* gene in all strains as a perfect hit (100% identity, 100% length coverage). *lmrD* encodes a subunit of the well-characterized LmrCD heterodimeric efflux ABC transporter, which is involved in multidrug resistance in *L. lactis* [24].

Glutamate decarboxylase was the only amino acid decarboxylase encoded in the genome of any strain. A few genes encoding housekeeping enzymes, multidrug efflux pumps, and proteins involved in heavy metal homeostasis and resistance were found in the genomes of the strains on the Rast Server. Except for a gene encoding a serine protease with partial similarity to the exfoliative toxin A from *Staphylococcus aureus* present in all *L. lactis* and *S. thermophilus* strains, no genes encoding toxins were identified.

## 3. Discussion

In the present work, four strains each of *L. lactis* and *S. thermophilus*, selected from among a set of GABA-producing LAB isolated from raw milk based on typing results, were phenotypically and genomically characterized. Genome analysis confirmed the identification of the strains at the species level (as *L. lactis* and *S. thermophilus*), and allocated two of the *L. lactis* strains (Lc 14.4 and Lc 21.1) to the biovar. diacetylactis. Both *L. lactis* and *S. thermophilus* are used worldwide as starters in cheese making and yogurt manufacturing [14]. Starters increase the availability of milk nutrients and contribute to generating appealing sensory properties in fermented dairy products [25]. GABA-producing LAB starters and adjunct cultures could be used in the development of functional fermented foods with a beneficial effect on consumer health [26]. However, the complete functional, technological, and safety characterization of such cultures is paramount. This can be accomplished by the combined phenotype testing and genome sequencing and analysis carried out in this work.

All eight analyzed strains converted MSG to GABA, both in laboratory media and in milk. The amount of GABA produced by the strains falls within the mid–high range compared to other *L. lactis* and *S. thermophilus* strains [10,15,27], but changes in manufacturing and/or ripening conditions may lead to improved GABA production in fermented dairy products [28]. The synthesis of GABA was sustained by the presence of operons harboring the genes encoding both the GAD enzyme (*gadB*) and the associated glutamate–GABA antiporter (*gadC*). The operon of *L. lactis* strains contained a gene (*gadR*) coding for a putative regulatory protein, while that of *S. thermophilus* strains did not. This is the usual gene content and gene order in these two LAB species [19,29]. Despite the assembly failure of the GABA operon in Lc 14.4, GABA production indicates that Lc 14.4 carries a fully functional biosynthetic pathway. Indeed, the strain was thought to carry GABA genes in both the chromosome and on a plasmid. Shuffling of genes between the bacterial chromosome and plasmids in *L. lactis* has been reported elsewhere [30]. The first role attributed to gene duplication is to provide enhanced enzymatic activity. Not surprisingly, Lc 14.4 showed the highest GABA production at 24 h. In contrast to the results in laboratory media and milk, none of the strains produced GABA in fecal homogenates. LAB strains with this property could have a role as probiotics, allowing GABA production in the gastrointestinal tract after ingestion [25]. This trait has been found in some lactobacilli and *Bifidobacterium* spp. strains of gut origin [31,32]. The presence of large numbers of bacteria well adapted to the gut environment may have prevented the dairy strains from growing (and thus producing GABA) in the fecal cultures.

The two strains of the biovar. diacetylactis proved to use citrate during growth in milk and produced noticeable amounts of acetoin. Both strains harbored a complete operon encoding all the components of the citrate lyase complex. However, the absence of *citP* in Lc 14.4, which utilized citrate from milk more efficiently than Lc 21.1, was surprising. It may be that one of the two citrate/H+ symporters in Lc 14.4, or both, which are inactivated by mutations in Lc 21.1 (a strain that harbors a citP gene), takes over the internalization of citrate. Although not yet studied in detail, citrate-utilizing (and diacetyl-producing) *L. lactis* strains without a recognizable CitP permease have been reported elsewhere [33,34]. None of the *S. thermophilus* strains utilized citrate from milk, but they all produced moderate amounts of acetoin. *S. thermophilus* is thought to produce diacetyl and acetoin by diverting the α-acetolactate that may result from an excess of branched-chain amino acids [35].

For rapid lactic acid production, and to attain a high cell density in milk, LAB require efficient systems for the utilization of lactose and caseins as a source of carbon and nitrogen [25]. The lactose operons found in the genomes of the sequenced strains corresponded to those described for *L. lactis* and *S. thermophilus*, respectively [21]. However, genes encoding extracellular proteinases with caseinolytic activity (PrtP and PrtS for *L. lactis* and *S. thermophilus*, respectively) were not identified in Lc 21.1, St 9.1, and St 21.1. These three strains grew slowly and did not coagulate milk after 24 h of incubation at their respective optimal temperatures. It is noteworthy that the PrtP sequence from *L. lactis* Lc 5.5 lacked a C-terminal region that included the final 163 amino acids. Deletions of the C-terminal part of the lactococcal caseinolytic protease retaining proteolytic activity have already been experimentally determined [36]. The *prtP* and *prtS* genes are either located on plasmids or on the chromosome. Chromosomal and plasmidic genes are thought to be acquired through horizontal transmission and can be mobilized through conjugation [37,38]. Surprisingly, the Lc 19.3 strain grew slowly in milk, even though it had a complete *prtP* gene. The reason for the slow growth in milk of this strain remains unknown. Changes in regulatory regions outside the coding regions may account for poor expression of the protease and/or its essential maturation partner protein in *L. lactis* (PrtM) [39]. The inclusion of protease-positive and protease-negative variants in the same starter mixture has been shown to reduce bitterness by certain *L. lactis* cultures [40], which suggests that due to the absence of caseinolytic activity, none of the strains should be excluded.

Although BAGEL 4 identified loci for bacteriocins and RiPPs in all strains, none inhibited the indicator strain. Some of these molecules (sactipeptides, streptides) have been identified in LAB only recently by mining their genomes with bioinformatic tools and may have narrow-spectrum activity, if any at all [41], which could explain the negative testing results. Further, classical bacteriocins may require specific target cells, functional quorum-sensing systems, or the presence of autoinducers that have to be provided by a helper strain [42]. Whatever the case, the absence of any strong antibacterial activity suggests compatibility between the strains.

Apart from the gene encoding the glutamate decarboxylase, none coding for amino acid decarboxylases were detected in any strain, which is consistent with the negative results obtained for the production of biogenic amines. Moreover, no transmissible resistance genes were identified, which agrees with the susceptibility of the strains to all relevant antibiotics tested. Further, despite the results of the BV-BRC annotation program, no genes coding for virulence factors or toxins were identified in the dedicated databases, except for those encoding exfoliative toxins. These proteins in LAB share significant structural homology with chymotrypsin, have very limited substrate specificity, and their physiological role remains to be determined [43]. All these data suggest that the examined strains are free of genes of concern, which indicates that they all are safe and can be confidently used in food systems.

The presence of prophages in a genome that, under uncertain environmental conditions, might be activated, thus inducing the ensuing lytic cycle, might impair a bacterial strainʼs use in industrial settings [44]. However, at least in *L. lactis* strains, the total absence of integrated phages in their genomes is highly unlikely [45].

## 4. Materials and Methods

### 4.1. Milk Sampling and Bacterial Culture Conditions

Twenty-four raw cow’s milk samples from different areas of the Asturias region (Northern Spain) were used as a source of *L. lactis* and *S. thermophilus* isolates. For the recovery of mesophilic microorganisms, milk dilutions in Ringer (Merck, Darmstadt, Germany) were plated on PCA agar (Merck) and incubated for 48–72 h at 32 °C. For the recovery of thermophiles, 40 mL milk samples were pasteurized at 63 °C for 30 min and incubated overnight at 42 °C. Dilutions of coagulated milk samples with good appearance and pleasant odor were then plated on M17 (Formedium, Norfolk, UK) agar supplemented with 0.5% glucose and 0.5% lactose (GLM17). The plates were incubated in an anaerobic jar with Anaerocult A (Merck) at 42 °C for 24–48 h. Five representative colonies produced at the two incubation temperatures were chosen at random and purified by subculturing on M17 agar supplemented with 1% glucose (GM17), or on GLM17, and incubated under the above conditions. Purified colonies (120 from each of the media) were inoculated in broth of their respective isolation media and incubated overnight. Cells were then centrifuged, suspended in fresh medium supplemented with 20% glycerol (*v*/*v*), and stored at −80 °C.

### 4.2. Identification and Typing

The overnight cultures of the isolates used to prepare the frozen stocks were also used for total genomic DNA extraction and purification using the QIAmp DNA Mini Kit (Qiagen, Hilden, Germany) following the manufacturer’s protocol for Gram-positive bacteria. Isolates were identified by amplification of their 16S rRNA genes with the universal primers 27F (5′-AGAGTTTGATCMTGGCTCAG-3′) and 1492R (5′-GGTTACCTTGTTACGACTT-3′). Amplification conditions, partial sequencing with the 27F primer, and comparison of the sequences with those in the databases were as reported previously by Rodríguez et al. [46].

GABA-producing *L. lactis* and *S. thermophilus* isolates were typed by combining the fingerprinting profiles obtained with primers BoxA2R (5′-ACGTGGTTTGAAGAGATTTTCG-3′), OPA18 (5′-AGGTGACCGT-3′), and M13 (5′-GAGGGTGGCGGTTCT-3′) following previously reported conditions [46]. Amplicons supplemented with EZ-Vision Two DNA Dye (VWR International) were separated in 2.5% agarose gels and visualized under UV light in a G-Box transilluminator (Syngene, Cambridge, UK), and the profiles obtained clustered using the Unweighted Pair Group Method with the Arithmetic mean algorithm (UPGMA) and Jaccard similarity coefficient.

### 4.3. GABA Production and Quantification

The ability of the *L. lactis* and *S. thermophilus* isolates to synthesize GABA from its precursor was tested, respectively, in GM17 or GLM17 broth media supplemented with 5 mM monosodium glutamate (MSG) after overnight incubation at either 32 °C (for *L. lactis*) or 42 °C (for *S. thermophilus*). GABA was detected and quantified by ultra-high-performance liquid chromatography (UHPLC) as previously reported [16]. Following the experience of previous works [16,47], isolates producing <0.64 mM of GABA in broth were considered non-producers.

### 4.4. Phenotypic, Technological, and Safety Profiles of GABA-Producing Strains

#### 4.4.1. Phenotype Characterization

The carbohydrate fermentation profiles of *L. lactis* and *S. thermophilus* strains were examined using the API-50 CHL system (bioMérieux, Montalieu-Vercieu, France). Enzyme activities were identified using the semi-quantitative API-ZYM system (bioMérieux). Urease activity in *S. thermophilus* was measured by assessing the release of ammonia from urea in a phenol red assay [48]. Briefly, 0.5 mL of an overnight culture in GLM17 was added to one volume of solution A (2 g of urea in 2 mL of ethanol and 4 mL of sterile water) and 19 volumes of solution B (KH_2_PO_4_ 1 g L^−1^; K_2_HPO_4_ 1 g L^−1^; NaCl 5 g L^−1^; phenol red 20 mg mL^−1^). This suspension was then incubated at 37 °C for up to 2 h; the development of a violet-red color indicated positive activity.

#### 4.4.2. Growth in Milk and Production of Organic Acids and Volatile Compounds

The *L. lactis* and *S. thermophilus* strains were grown in 10 mL of commercial UHT semi-skimmed milk (CAPSA, Siero, Spain) at 32 °C or 42 °C, respectively, for 12–24 h. Milk cultures were evaluated in triplicate for the production of organic acids, sugars, and volatile compounds (VOCs).

Organic acids and sugars from milk cultures were extracted and determined by UHPLC following the method of Alegría et al. [49]. Briefly, organic acids and sugars were separated using an ICSep ICE-ION-300 ion-exchange column (ThermoFisher Scientific, Waltham, MA, USA), employing an 8.5 mM H_2_SO_4_ aqueous mobile phase, an operating temperature of 65 °C, and a flow rate of 0.4 mL min^−1^. Organic acids were identified using a 996 Photodiode Array Detector (Waters, Milford, MA, USA) at 210 nm, and sugars using a 410 Differential Refractometer (Waters) at 280 nm, with quantification against calibration curves prepared with commercial standards.

The production of VOCs in milk was determined following the solid-phase microextraction (SPME)–gas chromatography method of Ziadi et al. [50]. Briefly, 5 mL of fermented milk was added to 20 mL screw-capped SPME vials (Agilent Technologies, Santa Clara, CA, USA). The vials were then sealed with a PTFE/silicone liner septum and equilibrated at 60 °C for 10 min with pulsed agitation for 5 s at 500 rpm using a PAL RSI 120 device (CTC Analytics, Zwingen, Switzerland). Volatile compounds were absorbed onto an ARR11-DVB-120/20 DVB/PDMS fiber (CTC Technologies) exposed to the headspace above the samples for 20 min at a depth of 40 mm (performed at 60 °C). The eluted compounds were identified based on their retention times and through comparison of their mass spectra with the Wiley Mass Spectral database (Wiley and Sons, NY, USA). Quantification was carried out through gas chromatography with a flame ionization detector (GC-FID) (HP5890 series II plus).

#### 4.4.3. GABA Production in Milk and Feces

GABA production in milk was assessed in commercial UHT semi-skimmed milk supplemented with 10 mM MSG after 12–24 h incubation at the optimal temperature for each strain. GABA was detected and quantified as above. In addition, GABA production by the strains was assessed in fecal homogenates supplemented with 30 mg mL⁻^1^ MSG following the procedure of Barrett et al. [31].

#### 4.4.4. Production of Biogenic Amines

The production of biogenic amines was examined in supernatants of the cultures following incubation at the optimal temperature for 48 h in GM17 or GLM17 supplemented with 5 mM of the precursor amino acids and amino acid-derived compounds (tyrosine, histidine, agmatine, and ornithine) (all from Sigma-Aldrich, Merck). As positive controls, the tyramine and histamine producers *Enterococcus faecalis* BA6 and *Lentilactobacillus parabuchneri* 2ES-1 were used. Processing and sample analysis were performed following the method described by Redruello et al. [47].

#### 4.4.5. Antibiotic Resistance

The antimicrobial resistance-susceptibility profile of the *L. lactis* and *S. thermophilus* strains in response to 16 antibiotics was assessed by broth microdilution using EULACBI1 and EULACBI2 plates (Trek Diagnostic System; ThermoFisher, Waltham, MASS, USA), as previously reported [46]. Minimum inhibitory concentration (MIC) values were compared to those reported by the European Food Safety Authority [51].

#### 4.4.6. Production of Bacteriocins

The production of bacteriocin-like substances was examined by an agar-well diffusion test using *Latilactobacillus sakei* CECT906T as an indicator, following previously reported inoculation procedures and incubation conditions [46]. Briefly, overnight supernatant cultures of the strains were neutralized, filter-sterilized, and added to wells made on MRS agar (Merck) plates inoculated (at 2%) with the indicator strain.

### 4.5. Whole-Genome Sequencing and Analysis

Next-generation sequencing libraries were prepared with total genomic DNA and using the TruSeq DNA PCR-free Sample Preparation Kit (Illumina, San Diego, CA, USA). Libraries were paired-end sequenced in a HiSeq 1500 System (Eurofins, Ebersberg, Germany). Genomes were annotated using the services of the Bacterial and Viral Bioinformatics Resource Center (BV-BRC; https://bv-brc.org/, accessed on 4 September 2023). For this, reads were first checked for quality with FastQC software, and contigs were assembled using the Unicycler program after comparing different variables using the Quality Assessment Tool for Genome Assemblies (QUAST). Errors were polished using Pilon and Racon software. Annotation with BV-BRC was performed using the RAST tool kit (RASTtk). Using BLAST tools (https://blast.ncbi.nlm.nih.gov/Blast.cgi, accessed on 5 September–5 December 2023), DNA and deduced protein sequences of interest were individually compared against those in the NCBI database. Whole-genome sequence data were used to confirm identifications through digital DNA–DNA hybridization (dDDH) and orthologous average nucleotide identity (orthoANI) analyses, as reported by Meier-Kolthoff and Göker [52] and Yoon et al. [53], respectively.

Genome sequences were searched for plasmids using PlasmidFinder (https://www.genomicepidemiology.org/services/, accessed on 18 September 2023). Ribosomally synthesized and post-translationally modified peptides and bacteriocins were sought using BAGEL 4 (http://bagel4.molgenrug.nl/, accessed on 19 September 2023). CRISPR-Cas systems were searched for using the CRISPRCasTyper v.1.6.4 online tool (https://crisprcastyper.crispr.dk, accessed on 19 September 2023), and Phaster software was used to search the genome sequence for prophages (https://phaster.ca, accessed on 20 September 2023).

Virulence, disease, and defense proteins were searched among the subsystems of the Rast Server (https://rast.nmpdr.org/, accessed on 5 September 2023), as well as in the Virulence Factors (VFDB; http://www.mgc.ac.cn/VFs/main.htm, accessed on 11 September 2023) and Victors (http://www.phidias.us/victors, accessed on 12 September 2023) databases.

## 5. Conclusions

Our genomic analysis of four GABA-producing strains each of *L. lactis* and *S. thermophilus* isolated from milk revealed the position and structure of the operon for the biosynthesis of this bioactive compound. In addition to laboratory media, all strains produced GABA in milk supplemented with MSG, but not in fecal homogenates. The genomic analysis determined the strains’ full biotechnological potential and contributed information guaranteeing their safety for use in food systems. Species-specific genes involved in growth in milk and the formation of flavor compounds (taste and aroma) were recorded, which agrees with the differential growth and the different profile of VOCs found for the strains of these two species. The phenotype and genome analyses indicated these strains to be suitable and safe to be used as starters or starter components for the manufacture of GABA-enriched functional fermented milks supplemented with MSG.

## Figures and Tables

**Figure 1 ijms-25-02328-f001:**
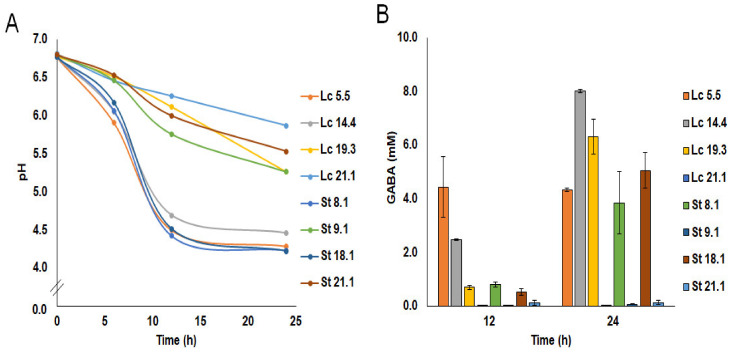
pH (**A**) and GABA production (**B**) of *L. lactis* (32 °C) and *S. thermophilus* (42 °C) strains in semi-skim milk supplemented with 10 mM monosodium glutamate (MSG).

**Figure 2 ijms-25-02328-f002:**
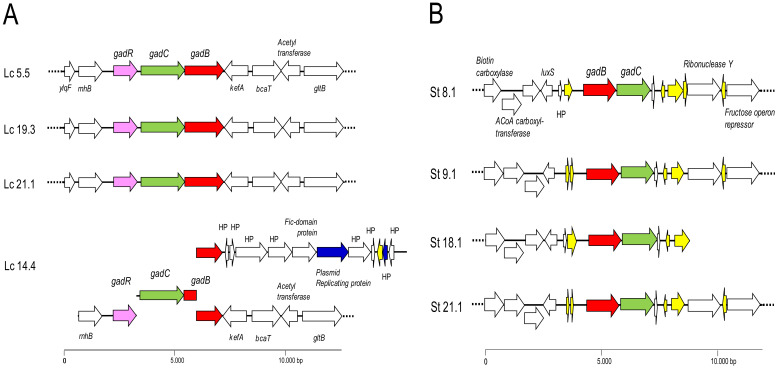
Genetic organization of the GABA operons found in the genome of *L. lactis* (**A**) and *S. thermophilus* (**B**) strains. Color code: in red, genes coding for the glutamate decarboxylase (*gadB*); in green, genes encoding the GABA/H+ antiporters (*gadC*); in purple, genes coding for the regulatory protein (*gadR*); in yellow, genes coding for transposases; in dark blue, genes encoding plasmid replication functions; in white, genes encoding proteins of other systems or ORFs encoding hypothetical proteins. The GABA operon of *L. lactis* Lc 14.4 was found to be spread over four different contigs (suggesting duplication of genes), of which one harbors genes encoding plasmid replication proteins. Dotted lines indicate that contigs extend beyond the depicted area.

**Table 1 ijms-25-02328-t001:** Sugar and organic acid profile of the GABA-producing *L. lactis* and *S. thermophilus* strains after growth in semi-skim milk.

Species/ Strain	Organic Acid or Sugar ^a^											
	Lactose	Glucose	Galactose	Orotic Acid	Citric Acid	Pyruvic Acid	Succinic Acid	Lactic Acid	Formic Acid	Acetic Acid	Uric Acid	Hippuric Acid
** *L. lactis* **												
Lc 5.5	6.68 ± 2.3	2.3 ± 0.1	44.4 ± 12.2	9.2 ± 2.5	272.5 ± 86.4	7.5 ± 1.7	2.1 ± 3.0	938.2 ± 224.4	7.3 ± 1.4	-	3.1 ± 1.0	-
Lc 14.4	4.49 ± 0.2	1.8 ± 0.1	51.4 ± 1.5	6.0 ± 0.4	-	7.4 ± 2.9	2.0 ± 2.8	826.3 ± 68.1	9.8 ± 3.3	73.7 ± 19.1	2.3 ± 0.1	2.3 ± 0.0
Lc 19.3	5.36 ± 1.2	1.9 ± 0.1	59.7 ± 22.0	9.4 ± 2.5	218.4 ± 47.7	2.8 ± 0.8	-	405.7 ± 132.6	-	6.6 ± 9.4	2.5 ± 0.7	0.6 ± 0.3
Lc 21.1	7.12 ± 2.5	2.5 ± 1.3	18.4 ± 5.8	10.7 ± 2.9	21.5 ± 8.3	15.1 ± 2.9	-	281.5 ± 62.9	-	74.2 ± 13.9	2.4 ± 0.6	3.2 ± 0.9
** *S. thermophilus* **												
St 8.1	5.05 ± 1.1	9.1 ± 2.1	982.4 ± 129.1	10.5 ± 1.1	263.2 ± 43.7	8.0 ± 0.3	-	1173.3 ± 55.5	-	-	2.9 ± 0.4	2.6 ± 0.5
St 9.1	5.84 ± 1.5	32.7 ± 7.4	391.9 ± 105.2	10.2 ± 2.6	251.4 ± 65.4	1.0 ± 0.1	-	464.0 ± 116.2	-	-	2.6 ± 0.7	2.1 ± 0.4
St 18.1	5.07 ± 3.0	9.0 ± 5.4	972.2 ± 547.8	10.8 ± 5.8	265.5 ± 147.3	8.3 ± 3.4	-	1203.6 ± 617.5	-	-	3.0 ± 1.7	2.6 ± 1.3
St 21.1	6.84 ± 2.3	77.0 ± 24	365.5 ± 105.8	11.2 ± 3.1	287.7 ± 92.2	1.6 ± 0.1	-	375.6 ± 90.4	-	-	2.6 ± 0.6	2.0 ± 0.5
**Milk (control)**	6.53 ± 0.3	42.2 ± 8.2	282.7 ± 152.4	8.5 ± 0.1	191.8 ± 1.4	1.4 ± 1.0	-	313.1 ± 184.2	-	-	1.8 ± 0.2	1.9 ± 0.3

^a^ Results expressed as mg per 100 mL of milk, except for lactose (g per 100 mL). -, not detected.

**Table 2 ijms-25-02328-t002:** Relative amounts of volatile compounds produced in semi-skim milk by *L. lactis* and *S. thermophilus* strains.

Compound ^a^	Control Milk	*L. lactis*				*S. thermophilus*			
		Lc 5.5	Lc 14.4	Lc 19.3	Lc 21.1	St 8.1	St 9.1	St 18.1	St 21.1
**Alcohols**									
2-Phenyl ethanol	-	19.13	33.26	-	-	-	-	-	-
Isoamyl alcohol	-	100.86	164.49	-	-	-	-	-	-
**Carboxylic acids**									
Acetic acid	-	3.63	14.36	3.28	6.20	1.39	0.84	1.63	0.64
Benzoic acid	-	13.44	0.86	12.26	0.57	2.55	5.19	0.95	4.24
Butanoic acid	0.26	4.43	3.85	4.67	2.39	6.09	5.09	6.38	4.89
Hexanoic acid	1.07	33.09	31.86	33.35	20.27	38.75	35.79	39.55	34.24
Heptanoic acid	-	0.98	1.01	1.04	-	0.85	0.81	0.94	0.81
Octanoic acid	1.87	35.99	37.83	36.51	30.90	41.25	40.64	42.48	39.72
n-Decanoic acid	2.62	20.69	21.14	19.67	18.07	21.60	20.34	22.58	20.23
9-Decenoic acid	-	2.27	2.37	2.32	1.82	2.80	2.54	2.80	2.47
Nonanoic acid	-	0.48	0.71	0.70	0.56	0.63	0.76	0.84	0.73
Dodecanoic acid	0.52	3.62	3.48	3.11	2.73	4.38	3.64	4.09	4.05
Tetradecanoic acid	-	-	-	-	-	1.04	0.82	0.95	0.88
Isovaleric acid	-	14.99	31.97	7.96	-	-	-	-	-
**Ketones**									
Acetoin^b^	-	-	17.02	-	25.54	13.58	8.31	12.81	9.82
2-Heptanone	2.67	-	-	-	-	3.86	3.26	4.42	3.38
2-Nonanone	1.85	2.12	1.90	2.05	3.06	2.56	2.60	2.96	2.52
2-Pentadecanone	-	-	0.55	-	-	-	-	-	-
2-Tridecanone	0.37	0.52	0.58	0.51	0.47	0.56	0.65	0.66	0.65
2-Undecanone	0.86	1.09	1.17	1.28	1.20	1.45	1.41	1.54	1.44
4-Octanone	-	40.70	79.50	20.59	-	-	-	-	-
2,5-Dimethyl-3-hexanone	-	-	3.99	0.67	-	-	-	-	-
**Lactones**									
δ-Decalactone	0.35	0.66	0.70	0.66	0.62	0.73	0.64	0.71	0.61
γ-Dodecalactone	0.35	0.45	0.43	0.40	0.40	0.47	0.66	0.48	0.50
δ-Dodecalactone	0.44	0.69	0.71	0.63	0.61	0.83	0.74	0.78	0.76
**Other**									
Benzaldehyde	0.60	1.49	1.63	1.35	0.72	0.77	-	0.81	-
Phenyl acetaldehyde	0.34	11.12	11.73	4.56	-	-	-	-	
2,3-Heptanedione	-	-	1.72	0.89	-	-	-	-	-
5-Methyl-2-phenyl-2-hexenal	-	1.45	1.62	-	-	-	-	-	-

^a^ Values of the relative area of the chromatograms as compared to that of the styrene internal standard.

**Table 3 ijms-25-02328-t003:** General features of the whole-genome sequence of GABA-producing *L. lactis* and *S. thermophilus* strains of this study.

Property/Protein-Encoding Genes	Species and Strains							
	*L. lactis*				*S. thermophilus*			
	Lc 5.5	Lc 14.4	Lc 19.3	Lc 21.1	St 8.1	St 9.1	St 18.1	St 21.1
Genome size (bp)	2,718,046	2,652,084	2,575,310	2,433,628	1,806,293	1,761,434	1,807,628	1,794,253
G + C content	35.17	35.16	35.17	34.79	39.77	39.40	39.89	39.47
No. of contigs	95	73	84	104	46	49	54	45
No. of coding sequences	2882	2762	2721	2558	2043	1989	2043	2041
No. of BV-BRC subsystems	207	207	208	204	202	202	202	202
Penicillin-binding proteins	2	2	2	2	2	2	2	2
Efflux-related proteins	28	24	29	27	27	34	28	34
Virulence Factors (VFDB/Victors)	2/8	2/9	1/8	1/7	1/39	1/41	1/39	1/41
Antibiotic Resistance (BV-BRC/CARD)	26/2	26/2	27/2	26/2	24/0	24/0	24/0	24/0
Resistance to heavy metals	1	2	2	1	2	2	2	2
rRNA sequences (23S + 16S + 5S)	4 (1 + 1 + 2)	4 (1 + 1 + 2)	4 (1 + 1 + 2)	4 (1 + 1 + 2)	3 (1 + 1 + 1)	2 (1 + 1 + 0)	3 (1 + 1 + 1)	2 (1 + 1 + 0)
tRNA molecules	52	50	50	51	37	30	35	30
Proteases	12	15	12	15	11	11	11	11
Peptidases	26	27	27	26	25	23	26	23
Transposases/integrases/excisionases	27	18	24	24	25	24	24	26
Phage-derived proteins	333	245	252	225	7	5	6	61
Plasmid replication proteins	13	11	14	12	1	1 (pUB110-type)	1	-
CRISPR-Cas loci	0	0	0	0	3	3	3	3
Bacteriocins	Lactococcin B and Q, sactipeptides	Enterolysin A, lactococcin A, B, and Q, sactipetides	2 x lactococcin B, linaridin	Lactococcin B, sactipeptides	BhtR, BlpD, BlpK, BlpU, streptide	BlpD, BlpK, BlpU, streptide	BlpD, BlpK, BlpU, streptide	BlpD, BlpK, BlpU, streptide

## Data Availability

The draft genome sequences of the strains examined in this study were deposited in GenBank under the BioSample accession numbers SAMN39094998, SAMN39094999, SAMN39095000, and SAMN39095001 (*L. lactis*), and SAMN39095002, SAMN39095003, SAMN39095004, and SAMN39095005 (*S. thermophilus*).

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
