# Peer review of "Phenotypic, Technological, Safety, and Genomic Profiles of Gamma-Aminobutyric Acid-Producing Lactococcus lactis and Streptococcus thermophilus Strains Isolated from Cow’s Milk"

_ijms, 2024, doi:10.3390/ijms25042328_

Round 1
Reviewer 1 Report
Comments and Suggestions for Authors
The work needs improvement

ok
Author Response
“Phenotypic, technological, safety, and genomic profiles of 2 GABA-producing Lactococcus lactis and Streptococcus thermophilus strains isolated from cowʼs milk”
- Abstract is good, but no findings data of the research paper was included in it therefore it should be included and General statement. should be cut short or deleted.
We partially agree with the Reviewer that the Abstract should be free of general statements and be centered on the key result. Therefore, we have deleted the first sentence and conveniently amended the second one. In addition, although we believe that the section already contained some results, we could have included the pivotal phenotypic traits whose genetic basis was discovered during genome analysis. These two parts now read as follows:
Gamma-aminobutyric acid (GABA)-producing lactic acid bacteria (LAB) can be used in the development of GABA-enriched functional fermented foods. In this work, four GABA-producing strains each of Lactococcus lactis and Streptococcus thermophilus were isolated from cowʼs milk, and their phenotypic, technological, and safety profiles were determined. Genome analysis provided genetic support for the majority of the analyzed traits; namely, GABA production, growth in milk, and absence of genes of concern.
- The introduction is written nicely but lacks the proper justification of present work, where is methodology adopted with reference to other studied or not should be added. Even though it is a survey work still needs proper Introduction.
In our opinion, the methodology is mostly well-explained and referenced in the corresponding section of Material and Methods. The location of this section at the end of the manuscript, as the Instructions to Authors of the journal requires, may cause some confusion among reviewers and readers costumed to different layouts.
As concerns the aim of the work, which was considered to be inadequately justified, we concluded that the Reviewer might be quite right. Therefore, it has been completely rewritten to make clear what the objective of our study was and the steps that to get there it includes. It now reads as:
Aiming at identifying GABA-producing starter candidates of L. lactis and S. thermophilus, a set of strains of these species isolated from raw milk were surveyed for GABA production. Among the higher producers, a set of eight strains (four L. lactis and four S. thermophilus) was subjected to a battery of phenotypic tests to determine their main technological and safety properties, followed by genome sequencing to reveal the basis of many of their key traits, including the synthesis of GABA. This manuscript reports on the phenotypic and genetic features that allowed us to propose the strains as conventional and functional starter cultures for dairy.
- In the manuscript material and method section the authors have taken 23 raw milk samples therefore what was the criteria procedure from selection.
Yes, not 23 but 24 raw milk samples were used as a source of mesophilic and thermophilic LAB species and strains. Five colonies from each of the media and temperature incubation conditions were initially gathered and stocked. From the 120 isolates of each mesophilic and thermophilic LAB, four were lost during purification (subculturing) or after freezing (during the recovery), then we ended up with 119 isolates from the GM17 (for L. lactis) and 117 from the GLM17 (for S. thermophilus) plates. For clarity, all these details have been added to either the Material and Methods or the Results sections.
- Section 4.2 no single reference is given; therefore, where the methodology has adopted from is not clear, add a reference.
The identification and typing methods were as previously reported by Rodríguez et al. [50]. In the previous section, the reference only applied to the second aspect (the typing). In this revised version, the text has been modified to include the identification method.
Amplification conditions, partial sequencing with the 27F primer, and comparison of the sequences with those in databases were previously reported by Rodríguez et al. [50].
GABA-producing L. lactis and S. thermophilus isolates were typed by combining the fingerprinting profiles obtained with primers BoxA2R (5ʼ-ACGTGGTTTGAAGAGATTTTCG-3ʼ), OPA18 (5ʼ-AGGTGACCGT-3ʼ) and M13 (5ʼ-GAGGGTGGCGGTTCT-3ʼ), following previously reported conditions [50].
- Line “The production of bacteriocin-like substances was examined by an agar-well diffusion test using Latilactobacillus sakei CECT906T as an indicator, following previously reported inoculation procedures and incubation conditions (Rodríguez et al., 2022).” Need to elaborate the procedure and author should check the reference style.
The well-assay procedure is now briefly explained, and the style of the reference modified; thank you for your remark.
The production of bacteriocin-like substances was examined by an agar-well diffusion test using Latilactobacillus sakei CECT906T as an indicator, following previously reported inoculation procedures and incubation conditions [50]. Briefly, overnight supernatant cultures of the strains were neutralized, filter-sterilized, and added to wells made on MRS agar (Merck) plates inoculated (at 2%) with the indicator strain.
- Figures: Authors have given a few graphs, some are not easy to understand from the point of view of readers also please explain the relevance of such tables and not in continuous form and very difficult.
Except for the figure in which the genes of the gad operons are depicted (Figure 2), all other graphs and “long” tables are included as supplementary materials. These are intended for comparative purposes with other works or researchers willing to delve into the genetic makeup of the strains of this study. Then, although we agree they are neither easy to understand nor of much relevance for this study, we would like to keep them.
- Statistical analysis is missing by the author be elaborated it should be added in revised manuscript.
Beyond standard deviations, which are already indicated in tables and figures, statistical analyses, such as ANOVA or any other comparative tests, are not of relevance in this study, because all measured traits (GABA production, production of organic acids, etc.) are reported descriptively. Of course, differences between strains for each of the parameters were found, but there are not any conditions or strains of reference to which the results can be compared to.
- As per the results findings the Conclusion given is a bit lengthy, it should be a complete gist of study in bullet points which is easy to understand for the viewers.
The section of reference has now been shortened, leaving only the key points that were addressed in the current study. Aspects that were most closely to the discussion (speculative) or redundant have been deleted. However, we still would like to maintain the structure of a simple paragraph, as recommended the Instructions by Authors of the journal. Bullet points might be more appropriate to sections such as the Highlights.
Conclusions
Genomic analysis of four GABA-producing strains each of L. lactis and S. thermophilus isolated from milk, revealed the position and structure of the operon for the biosynthesis of this bioactive compound. In addition to laboratory media, all strains produced GABA in milk supplemented with MSG, but not in fecal homogenates. Genomic analysis determined the strainsʼ full biotechnological potential and contributed information guaranteeing their safety for use in food systems. Species-specific genes involved in growing in milk and the formation of flavor compounds (taste and aroma) were recorded, which agrees with the differential growth and the different profile of VOCs found for the strains of the two species. Phenotype and genome analyses indicate the strains to be suitable and safe to be used as starter or starter components for the manufacture of GABA-enriched functional fermented milks supplemented with MSG.
Reviewer 2 Report
Comments and Suggestions for Authors
Dear authors,
The manuscript by Valenzuela and colleagues presents a systematic procedure for GABA-producing strains screening, identification, charaterisation and safety analysis. I see the work is of useful contribution for the food industry. However, few questions must be clear.
1. In line 98, "Isolates producing <0.64 mM of GABA in broth were considered non-producers." Could you kindly clarify why you chose 0.64mM as a limit? On the other hand, Fig 1-B showed the GABA-producing yield of 8 strains; two strains (Lc21.1 and St9.1) may generate GABA at less than 0.64 mM; if so, why did you choose those two strains for further investigation?
2. The results in the Supplementary Part show that Lc5.5, Lc19.3, St9.1, and St21.1 have similar metabiolize and genetic profiles. However, why do they possess different growth capacities as shown in Fig. 1?
3. In the Supplementary Part, I observe that Lc16.6, Lc21, St20.1, St1.1, St16.2, and St8.1 were added as extras in Fig. S1. please explain it.
4. In line 270, please include a reference to the text "Three lactococcal strains showed a locus for the synthesis of sactipeptides (sulfur-to-α-carbon-containing peptides also classified as Class Ic bacte riocins)."
5. Latilactobacillus sakei CECT906T was used as indicator to assess the bacteriocin activity of eight strains in your manuscript. And “none inhibited the highly susceptible strain used as an indicator” was mentioned in line 361 in your manuscript. Bacteriocins are ribosomlly synthesized antimicrobial peptides, and normally showed inhibitory activity against close related specise (Cotter PD, Ross RP, Hill C. Bacteriocins-a viable alternative to antibiotics? Nat Rev Microbiol. 2013,11(2):95-105. Diep DB, Nes IF. Ribosomally synthesized antibacterial peptides in Gram positive bacteria. Curr Drug Targets. 2002,3(2):107-22. doi: 10.2174/1389450024605409.). According to your findings, Lactococcus spp. and Strepotoccus spp. exhibit minimal antibacterial effect against Latilactobacillus sakei. The "highly susceptible strain" probably needs to be modified.
6. Some errors in lines 67, 308, 428, and so on should be double-checked and corrected.
Author Response
The manuscript by Valenzuela and colleagues presents a systematic procedure for GABA-producing strains screening, identification, charaterisation and safety analysis. I see the work is of useful contribution for the food industry. However, few questions must be clear.
- In line 98, "Isolates producing <0.64 mM of GABA in broth were considered non-producers." Could you kindly clarify why you chose 0.64 mM as a limit? On the other hand, Fig 1-B showed the GABA-producing yield of 8 strains; two strains (Lc21.1 and St9.1) may generate GABA at less than 0.64 mM; if so, why did you choose those two strains for further investigation?
The testing medium contained already a certain amount of GABA (0.3 mM). It also contains some quantities of the free amino acid glutamate, of which some can be transformed into GABA by unspecific amino acid decarboxylase activities. In our Institute, there has been a lot of research on GABA production. The 0.64 mM threshold has been established with data from previous works, of which some are referenced in the manuscript (Valenzuela et al., 2019; Redruello et al., 2021). The sentence with the threshold has been transferred to the corresponding heading of the Material and Method section (where, in fact, it previously was, and from where it was advanced because of the position of Material and Methods in the journal).
Following the experience of previous works [19, 51], isolates producing <0.64 mM of GABA in broth were considered non-producers.
- The results in the Supplementary Part show that Lc5.5, Lc19.3, St9.1, and St21.1 have similar metabiolize and genetic profiles. However, why do they possess different growth capacities as shown in Fig. 1?
Yes, you are quite right; the strains of each L. lactis and S. thermophilus species are quite similar. The different growth capacity and GABA production in milk of the strains reported in Fig. 1, however, result from either the presence or absence of only a key gene: that encoding the essential caseinolytic proteinase. Although we believe this was already suggested into the text, we have modified the sentence to stress this fact, as follows:
GABA production in supplemented milk was primarily associated with growth in this medium (Fig. 1); strains that did not grow were not able to synthesize GABA in milk.
- In the Supplementary Part, I observe that Lc16.6, Lc21, St20.1, St1.1, St16.2, and St8.1 were added as extras in Fig. S1. please explain it.
These corresponded to the six extra strains that were initially found to produce GABA but were not selected for further study. These strains included isolates considered to be replicates of the same strains because they shared a homology higher than the repeatability of the study (93%), which is the case for non-tested S. thermophilus isolates, or for our limited financial capacity that did not allow us to sequence more strains at that time, which is the case for non-tested all L. lactis strains. Although these reasons are not stated there, we have modified the footnote of Fig. S2 to clearly explain which strains were selected for phenotypic analyses and genome sequencing. Nevertheless, as indicated in the text, the strains selected recovered the largest genetic diversity found by the typing technique among the L. lactis isolates.
In bold, selected strains for further phenotypic testing and genome sequencing.
Among those, four genetically unrelated strains each of L. lactis (Lc 5.5, Lc 14.4, Lc 19.3, and Lc 21.1) and S. thermophilus (St 8.1, St 9.1, St 18.1, and St 21.1) were selected for the characterization of relevant technological and safety properties, as well as for genome sequencing and analysis.
- In line 270, please include a reference to the text "Three lactococcal strains showed a locus for the synthesis of sactipeptides (sulfur-to-α-carbon-containing peptides also classified as Class Ic bacte riocins)."
This sentence does not need a reference because the results correspond to those obtained in this work. To make it pretty clear, Table 3 was added at the end of the sentence, as follows:
Three lactococcal strains showed a locus for the synthesis of sactipeptides (sulfur-to-α-carbon-containing peptides; also classified as Class Ic bacteriocins) (Table 3).
- Latilactobacillus sakei CECT906T was used as an indicator to assess the bacteriocin activity of eight strains in your manuscript. And “none inhibited the highly susceptible strain used as an indicator” was mentioned in line 361 in your manuscript. Bacteriocins are ribosomally synthesized antimicrobial peptides, and normally showed inhibitory activity against closely related species (Cotter PD, Ross RP, Hill C. Bacteriocins-a viable alternative to antibiotics? Nat Rev Microbiol. 2013,11(2):95-105. Diep DB, Nes IF. Ribosomally synthesized antibacterial peptides in Gram positive bacteria. Curr Drug Targets. 2002,3(2):107-22. doi: 10.2174/1389450024605409.). According to your findings, Lactococcus spp. and Strepotoccus spp. exhibit minimal antibacterial effect against Latilactobacillus sakei. The "highly susceptible strain" probably needs to be modified.
Yes, it may seem so. However, the CECT 916T strain has been used in many different works in our laboratory since the 90s. With this strain, we have detected bacteriocins from species of Lactococcus (lactococcin 972), Enterococcus (several enterocins) and Lactobacillus (plantaricin C). We thought it wouldn´t add anything to the discussion referring to all these works; tha´s why they do not appear there. Nonetheless, the expression was deleted.
The absence of antimicrobial activity was discussed on the basis of other possibilities (see below that part in the Discussion section). Sactipeptides and streptides from LAB have been identified recently by bioinformatics. None of the strains having genetic potential to synthesize these molecules has ever been reported to show inhibitory activity by these specific peptides. Further, the production of some bacteriocins relies on autoinducers produced by helper strains, as is the case for the production of some plantaricins as referenced in the text. Nonetheless, the assay was only performed to get some insights into the compatibility of the strains. This requires the absence of (strong) antimicrobial activities that could unbalance the components of complex starter mixtures. This fact has now been added to the paragraph.
Although BAGEL 4 identified loci for bacteriocins and RiPPs in all strains, none inhibited the indicator strain used. Some of these molecules (sactipeptides, streptides) have been identified in LAB only recently by mining their genomes with bioinformatic tools and may have if at all, a narrow-spectrum activity [45], which could explain the negative testing results. Further, classical bacteriocins may require specific target cells, functional quorum sensing systems, or the presence of autoinducers that have to be provided by a helper strain [46]. Whatever the case, the absence of any strong antibacterial activity suggests compatibility between the strains.
- Some errors in lines 67, 308, 428, and so on should be double-checked and corrected.
Thanks for noting; the whole text has been reviewed and, wherever identified, typing errors and grammatical mistakes corrected.
Reviewer 3 Report
Comments and Suggestions for Authors
“What was the letter-number strain designation of the bacteria used in the research? This letter-number designation should be in the title, abstract, conclusion, title of the figures, and in the text of the Manuscript. Proper identification of the bacteria used as a probiotic is very important because the properties, and functional characteristics of bacteria are strain-dependent and cannot be extrapolated to species or genus.”
Please explain in the introduction more clearly why the research study is important.
“-The manuscript abounds in many details but it is not clear the last objective of the study, which should appear clearly in the abstract and the introduction.”
-Please update references from 10 to 15 years at oldest.
-please improve the discuss. Please explain the implications of the results.
“Improve Conclusions according to everything observed, clearly indicating if the objectives of the work were met and if the results have possible applications”
Author Response
- What was the letter-number strain designation of the bacteria used in the research? This letter-number designation should be in the title, abstract, conclusion, title of the figures, and in the text of the Manuscript. Proper identification of the bacteria used as a probiotic is very important because the properties, and functional characteristics of bacteria are strain-dependent and cannot be extrapolated to species or genus.
I fully agree with the reviewer that strains intended to be used as probiotics should be identified at the strain level. However, the strains characterized in this work are considered dairy starter candidates. In this context, and due to the presence of lytic bacteriophages in the milk environment, mixtures of strains are usually employed rather than individual strains. Therefore, we believe it is not worth stating the code of the strains in the title or other sections, such as Abstract and Conclusions.
- Please explain in the introduction more clearly why the research study is important.
In the Introduction section, the aim of the work has been completely rewritten in an attempt to clarify and stress the aim and impact of the work. Hope the importance of the work is clearly stated in the revised version of the manuscript.
Aiming at identifying GABA-producing starter candidates of L. lactis and S. thermophilus, a set of strains of these species isolated from raw milk were surveyed for GABA production. Among the higher producers, a set of eight strains (four L. lactis and four S. thermophilus) was subjected to a battery of phenotypic tests to determine their main technological and safety properties, followed by genome sequencing to reveal the basis of many of their key traits, including GABA production. This manuscript reports on the phenotypic and genetic features that allowed us to propose the strains as conventional and functional starter cultures for dairy.
- The manuscript abounds in many details but it is not clear the last objective of the study, which should appear clearly in the abstract and the introduction.
In line with the above response, the Abstract section has also been modified to leave out the general statement on GABA and health and to focus the text on the objective of the study. This initial sentence is linked with the final conclusion in the same section; see below:
Gamma-aminobutyric acid (GABA)-producing lactic acid bacteria (LAB) can be used as starters in the development of GABA-enriched functional fermented foods.
Altogether these results suggest all eight strains may be considered candidates for use as starter or components of mixed LAB cultures for the manufacture of GABA-enriched fermented dairy products.
- Please update references from 10 to 15 years at oldest.
It may feel that there are some old references, and they are, some of these references are the only works on the topic, such as Juillard et al. 1996, Bruinenberg et al., 2000, Nardi et al., 1997, 1997; Higuchi et al., 1997; Hayes et al., 1990, Vos et al., 1989, or include pioneering works in certain topics that we would like to acknowledge (Nomura et al., 1998; Nomura et al., 2000; Sanders et al., 1998; Desiere et al., 2002). Nonetheless, during the revision we have reviewed all the references and deleted four that were considered redundant or not appropriate. Hope, this action, at least partially, meets your requirement.
- please improve the discuss. Please explain the implications of the results.
Inevitably, we think we have to discuss in the context of the published literature both the phenotypic and genetic aspects of this work. However, as all three reviewers stressed the feeling that neither the aim of the work nor the importance of the results have been well explained, we can conclude that these aspects were not clear in the previous version. This prompted us to fully rewrite and reordered the first paragraph of the Discussion. Together with the new paragraph in the Introduction and the whole new Conclusions section, we hope to have addressed this weakness. The mentioned new paragraph reads now as follows:
In the present work, four strains each of L. lactis and S. thermophilus, selected from among a set of GABA-producing LAB isolated from raw milk based on the typing results, were phenotypically and genomically characterized. Genome analysis confirmed the identification of the strains at the species level (as L. lactis and S. thermophilus), and allocated two of the L. lactis strains (Lc 14.4 and Lc 21.1) to the biovar. diacetylactis. Both L. lactis and S. thermophilus are used worldwide as starters in cheese making and yogurt manufacturing [14]. Starters increase the availability of milk nutrients and contribute to generating appealing sensory properties in fermented dairy products [25]. GABA-producing LAB starter and adjunct cultures could be used in the development of functional fermented foods with a beneficial effect on consumer health [26]. However, a complete functional, technological, and safety characterization of such cultures is paramount. This can be accomplished by the combined phenotype testing and genome sequencing and analysis carried out in this work.
- Improve Conclusions according to everything observed, clearly indicating if the objectives of the work were met and if the results have possible applications.
Following the suggestion of Reviewer 1 and yours, the Conclusions section has been shortened. Redundancies, discussion and prospects have now been left out, and the conclusion focuses directly on the main findings and possible applications.
Genomic analysis of four GABA-producing strains each of L. lactis and S. thermophilus isolated from milk, revealed the position and structure of the operon for the biosynthesis of this bioactive compound. In addition to laboratory media, all strains produced GABA in milk supplemented with MSG, but not in fecal homogenates. Genomic analysis determined the strainsʼ full biotechnological potential and contributed information guaranteeing their safety for use in food systems. Species-specific genes involved in growing in milk and the formation of flavor compounds (taste and aroma) were recorded, which agrees with the differential growth and the different profile of VOCs found for the strains of the two species. Phenotype and genome analyses indicate the strains to be suitable and safe to be used as starter or starter components for the manufacture of GABA-enriched functional fermented milks supplemented with MSG.